# A Communication-Efficient Distributed Gradient Clipping Algorithm for Training Deep Neural Networks

**Mingrui Liu**[1*]**, Zhenxun Zhuang**[2]**, Yunwen Lei**[3]**, Chunyang Liao**[4]
[1]Department of Computer Science, George Mason University, Fairfax, VA 22030, USA
[2]Meta Platforms, Inc., Bellevue, WA, 98004, USA
[3] School of Computer Science, University of Birmingham, United Kingdom
[4] Department of Mathematics, Texas A&M University, College Station, Texas 77840, USA
`mingruil@gmu.edu, oldboymls@gmail.com, yunwen.lei@hotmail.com`

## Abstract

In distributed training of deep neural networks, people usually run Stochastic Gradient Descent (SGD) or its variants on each machine and communicate with other machines periodically. However, SGD might converge slowly in training some deep neural networks (e.g., RNN, LSTM) because of the exploding gradient issue. Gradient clipping is usually employed to address this issue in the single machine setting, but exploring this technique in the distributed setting is still in its infancy: it remains mysterious whether the gradient clipping scheme can take advantage of multiple machines to enjoy parallel speedup. The main technical difficulty lies in dealing with nonconvex loss function, non-Lipschitz continuous gradient, and skipping communication rounds simultaneously. In this paper, we explore a relaxed-smoothness assumption of the loss landscape which LSTM was shown to satisfy in previous works, and design a communication-efficient gradient clipping algorithm. This algorithm can be run on multiple machines, where each machine employs a gradient clipping scheme and communicate with other machines after multiple steps of gradient-based updates. Our algorithm is proved to have $O\left(\frac{1}{N\epsilon^4}\right)$ iteration complexity and $O(\frac{1}{\epsilon^3})$ communication complexity for finding an $\epsilon$-stationary point in the homogeneous data setting, where $N$ is the number of machines. This indicates that our algorithm enjoys linear speedup and reduced communication rounds. Our proof relies on novel analysis techniques of estimating truncated random variables, which we believe are of independent interest. Our experiments on several benchmark datasets and various scenarios demonstrate that our algorithm indeed exhibits fast convergence speed in practice and thus validates our theory.

## 1   Introduction

Deep learning has achieved tremendous successes in many domains including computer vision [23, 15], natural language processing [5] and game [39]. To obtain good empirical performance, people usually need to train large models on a huge amount of data, and it is usually very computationally expensive. To speed up the training process, distributed training becomes indispensable [4]. For example, Goyal et al. [12] trained a ResNet-50 on ImageNet dataset by distributed SGD with minibatch size 8192 on 256 GPUs in only one hour, which not only matches the small minibatch

---

*Corresponding Author: Mingrui Liu (`mingruil@gmu.edu`). The code is available at `https://github.com/MingruiLiu-ML-Lab/Communication-Efficient-Local-Gradient-Clipping`

36th Conference on Neural Information Processing Systems (NeurIPS 2022).

Table 1: Comparison of Iteration and Communication Complexity of Different Algorithms for finding a point whose gradient's magnitude is smaller than $\epsilon$ (i.e., $\epsilon$-stationary point defined in Definition 3), $N$ is the number of machines, the meaning of other constants can be found in Assumption 1. For the complexity of [9] in this table, we assume the gradient norm is upper bounded by $M$ such that the gradient is $(L_0 + L_1 M)$-Lipschitz. However, the original paper of [9] does not require bounded gradient; instead, they require $L$-Lipschitz gradient and bounded variance $\sigma^2$. Under their assumption, their complexity result is $O\left(\Delta L \epsilon^{-2} + \Delta L \sigma^2 \epsilon^{-4}\right)$.

| Algorithm | Setting | Iteration Complexity | Communication Complexity |
|---|---|---|---|
| SGD [9] | Single | $O\left(\Delta(L_0 + L_1 M)\epsilon^{-2} + \Delta(L_0 + L_1 M)\sigma^2 \epsilon^{-4}\right)$ | N/A |
| Clipped SGD [51] | Single | $O\left((\Delta + (L_0 + L_1\sigma)\sigma^2 + \sigma L_0^2/L_1)^2)\epsilon^{-4}\right)$ | N/A |
| Clipping Framework [50] | Single | $O\left(\Delta L_0 \sigma^2 \epsilon^{-4}\right)$ | N/A |
| Naive Parallel of [50] | Distributed | $O\left(\Delta L_0 \sigma^2/(N\epsilon^4)\right)$ | $O\left(\Delta L_0 \sigma^2/(N\epsilon^4)\right)$ |
| Ours (this work) | Distributed | $O(\Delta L_0 \sigma^2/(N\epsilon^4))$ | $O\left(\Delta L_0 \sigma \epsilon^{-3}\right)$ |

accuracy but also enjoys parallel speedup, and hence improves the running time. Recently, local SGD [40, 47], as a variant of distributed SGD, achieved tremendous attention in federated learning community [29]. The local SGD algorithm runs multiple steps of SGD on each machine before communicating with other clients.

Despite the empirical success of distributed SGD and its variants (e.g., local SGD) in deep learning, they may not exhibit good performance when training some neural networks (e.g., Recurrent Neural Networks, LSTMs), due to the exploding gradient problem [33, 34]. To address this issue, Pascanu et al. [34] proposed to use the gradient clipping strategy, and it has become a standard technique when training language models [8, 35, 31]. There are some recent works trying to theoretically explain gradient clipping from the perspective of non-convex optimization [51, 50]. These works are built upon an important observation made in [51]: for certain neural networks such as LSTM, the gradient does not vary uniformly over the loss landscape (i.e., the gradient is not Lipschitz continuous with a uniform constant), and the gradient Lipschitz constant can scale linearly with respect to the gradient norm. This is referred to as the relaxed smoothness condition (i.e., $(L_0, L_1)$-smoothness defined in Definition 2), which generalizes but strictly relaxes the usual smoothness condition (i.e., $L$-smoothness defined in Definition 1). Under the relaxed smoothness condition, Zhang et al. [51, 50] proved that gradient clipping enjoys polynomial-time iteration complexity for finding the first-order stationary point in the single machine setting, and it can be arbitrarily faster than fix-step gradient descent. In practice, both distributed learning and gradient clipping are important techniques to accelerate neural network training. However, the theoretical analysis of gradient clipping is only restricted to the single machine setting [51, 50]. Hence it naturally motivates us to consider the following question:

**Is it possible that the gradient clipping scheme can take advantage of multiple machines to enjoy parallel speedup in training deep neural networks, with only infrequent communication?**

In this paper, we give an affirmative answer to the above question. Built upon the relaxed smoothness condition as in [51, 50], we design a communication-efficient distributed gradient clipping algorithm. The key characteristics of our algorithm are: (i) unlike naive parallel gradient clipping algorithm [2] which requires averaging model weights and gradients from all machines for every iteration, our algorithm only aggregates weights with other machines after a certain number of local updates on each machine; (ii) our algorithm clips the gradient according to the norm of the local gradient on each machine, instead of the norm of the averaged gradients across machines as in the naive parallel algorithm. These key features make our algorithm amenable to the distributed setting with infrequent communication, and it is nontrivial to establish desired theoretical guarantees (e.g., linear speedup, reduced communication complexity). The main difficulty in the analysis lies in dealing with the nonconvex objective function, non-Lipschitz continuous gradient, and skipping communication rounds simultaneously. Our main contribution is summarized as the following:

---

[2]Naive Parallel of [50] stands for the algorithm which clips the gradient based on the globally averaged gradient at every iteration with constant minibatch size.

- We design a novel communication-efficient distributed stochastic local gradient clipping algorithm, namely CELGC, for solving a nonconvex optimization problem under the relaxed smoothness condition. The algorithm only needs to clip the gradient according to the local gradient's magnitude and globally averages the weights on all machines periodically. To the best of our knowledge, this is the first work proposing communication-efficient distributed stochastic gradient clipping algorithms under the relaxed smoothness condition.

- Under the relaxed smoothness condition, we prove iteration and communication complexity results of our algorithm for finding an $\epsilon$-stationary point, when each machine has access to the same data distribution. First, comparing with [50], we prove that our algorithm enjoys linear speedup, which means that the iteration complexity of our algorithm is reduced by a factor of $N$ (the number of machines). Second, comparing with the naive parallel version of the algorithm of [50], we prove that our algorithm enjoys better communication complexity. The detailed comparison over existing algorithms under the same relaxed smoothness condition is described in Table 1. To achieve this result, we introduce a novel technique of estimating truncated random variables, which is of independent interest and could be applied in related problems involving truncation operations such as gradient clipping.

- We empirically verify our theoretical results by conducting experiments on different neural network architectures on benchmark datasets. The experimental results demonstrate that our proposed algorithm indeed exhibits speedup in practice.

## 2  Related Work

**Gradient Clipping/Normalization Algorithms**   In deep learning literature, gradient clipping (normalization) technique was initially proposed by [34] to address the issue of exploding gradient problem in [33], and it has become a standard technique when training language models [8, 35, 31].It is shown that gradient clipping is robust and can mitigate label noise [30]. Recently gradient normalization techniques [45, 46] were applied to train deep neural networks on the very large batch setting. For example, You et al. [45] designed the LARS algorithm to train a ResNet50 on ImageNet with batch size 32k, which utilized different learning rates according to the norm of the weights and the norm of the gradient. In optimization literature, gradient clipping (normalization) was used in early days in the field of convex optimization [7, 1, 38]. Nesterov [32] and Hazan et al. [14] considered normalized gradient descent for quasi-convex functions in deterministic and stochastic cases respectively. Gorbunov et al. [10] designed an accelerated gradient clipping method to solve convex optimization problems with heavy-tailed noise in stochastic gradients. Mai and Johansson [26] established the stability and convergence of stochastic gradient clipping algorithms for convex and weakly convex functions. In nonconvex optimization, Levy [24] showed that normalized gradient descent can escape from saddle points. Cutosky and Mehta [3] showed that adding a momentum provably improves the normalized SGD in nonconvex optimization. Zhang et al. [51] and Zhang et al. [50] analyzed the gradient clipping for nonconvex optimization under the relaxed smoothness condition rather than the traditional $L$-smoothness condition in nonconvex optimization [9]. However, all of them only consider the algorithm in the single machine setting or the naive parallel setting, and none of them can apply to the distributed setting where only limited communication is allowed.

**Communication-Efficient Algorithms in Distributed and Federated Learning**   In large-scale machine learning, people usually train their model using first-order methods on multiple machines and these machines communicate and aggregate their model parameters periodically. When the function is convex, there is a scheme named one-shot averaging [56, 28, 54, 37], in which every machine runs a stochastic approximation algorithm and averages the model weights across machines only at the very last iteration. The one-shot averaging scheme is communication-efficient and enjoys statistical convergence with one pass of the data [54, 37, 16, 22], but the training error may not converge in practice. Mcmahan et al. [29] considered the Federated Learning setting where the data is decentralized and might be non-i.i.d. across devices and communication is expensive and designed the very first algorithm for federated learning (a.k.a., FedAvg). Stich [40] considered a concrete case of FedAvg, namely local SGD, which runs SGD independently in parallel on different works and averages the model parameters only once in a while. Stich [40] also showed that local SGD enjoys linear speedup for strongly-convex objective functions. There are also some works analyzing local SGD and its variants on convex [6, 20, 19, 43, 44, 11, 49] and nonconvex smooth functions [55, 47, 48, 17, 41, 25, 2, 13, 19]. Recently, Woodworth et al. [43, 44] analyzed advantages

and drawbacks of local SGD compared with minibatch SGD for convex objectives. Woodworth et al. [42] proved hardness results for distributed stochastic convex optimization. Reddi et al. [36] introduced a general framework of federated optimization and designed several federated versions of adaptive optimizers. Zhang et al. [52] considered employing gradient clipping to optimize $L$-smooth functions and achieve differential privacy. Koloskova et al. [21] developed a unified theory of decentralized SGD with changing topology and local updates for smooth functions. Zhang et al. [53] developed a federated learning framework for nonconvex smooth functions for non-i.i.d. data. Due to a vast amount of literature on federated learning and limited space, we refer readers to [18] and references therein. However, all of these works either assume the objective function is convex or $L$-smooth. To the best of our knowledge, our algorithm is the first communication-efficient algorithm that does not rely on these assumptions but still enjoys linear speedup.

## 3 Preliminaries, Notations and Problem Setup

**Preliminaries and Notations**   Denote $\|\cdot\|$ by the Euclidean norm. We denote $f : \mathbb{R}^d \to \mathbb{R}$ as the overall loss function, and $F : \mathbb{R}^d \to \mathbb{R}$ as the loss function on $i$-th machine, where $i \in [N] := \{1, \ldots, N\}$. Denote $\nabla h(\mathbf{x})$ as the gradient of $h$ evaluated at the point $\mathbf{x}$, and denote $\nabla h(\mathbf{x}; \xi)$ as the stochastic gradient of $h$ calculated based on sample $\xi$.

**Definition 1** ($L$-smoothness). *A function $h$ is $L$-smooth if $\|\nabla h(\mathbf{x}) - \nabla h(\mathbf{y})\| \leq L\|\mathbf{x} - \mathbf{y}\|$ for all $\mathbf{x}, \mathbf{y} \in \mathbb{R}^d$.*

**Definition 2** (($L_0, L_1$)-smoothness). *A second order differentiable function $h$ is $(L_0, L_1)$-smooth if $\|\nabla^2 h(\mathbf{x})\| \leq L_0 + L_1\|\nabla h(\mathbf{x})\|$ for any $\mathbf{x} \in \mathbb{R}^d$.*

**Definition 3** ($\epsilon$-stationary point). *$\mathbf{x} \in \mathbb{R}^d$ is an $\epsilon$-stationary point of the function $h$ if $\|\nabla h(\mathbf{x})\| \leq \epsilon$.*

**Remark:**   $(L_0, L_1)$-smoothness is strictly weaker than $L$-smoothness. First, we know that $L$-smooth functions is $(L_0, L_1)$-smooth with $L_0 = L$ and $L_1 = 0$. However the reverse is not true. For example, consider the function $h(x) = x^4$, we know that the gradient is not Lipschitz continuous and hence $h$ is not $L$-smooth, but $|h''(x)| = 12x^2 \leq 12 + 3 \times 4|x|^3 = 12 + 3|h'(x)|$, so $h(x) = x^4$ is $(12, 3)$-smooth. Zhang et al. [51] empirically showed that the $(L_0, L_1)$-smoothness holds for the AWD-LSTM [31]. In nonconvex optimization literature [9, 50], the goal is to find an $\epsilon$-stationary point since it is NP-hard to find a global optimal solution for a general nonconvex function.

**Problem Setup**   In this paper, we consider the following optimization problem using $N$ machines:
$$\min_{\mathbf{x} \in \mathbb{R}^d} f(\mathbf{x}) = \mathbb{E}_{\xi \sim \mathcal{D}} \left[ F(\mathbf{x}; \xi) \right], \tag{1}$$
where $\mathcal{D}$ stands for the data distribution which each machine has access to, and $f$ is the population loss function.

We make the following assumptions throughout the paper.

**Assumption 1.**      *(i) $f(\mathbf{x})$ is $(L_0, L_1)$-smooth, i.e., $\|\nabla^2 f(\mathbf{x})\| \leq L_0 + L_1\|\nabla f(\mathbf{x})\|$, for $\forall \mathbf{x} \in \mathbb{R}^d$.*

     *(ii) There exists $\Delta > 0$ such that $f(\mathbf{x}_0) - f_* \leq \Delta$, where $f_*$ is the global optimal value of $f$.*

     *(iii) For all $\mathbf{x} \in \mathbb{R}^d$, $\mathbb{E}_{\xi \sim \mathcal{D}} [\nabla F(\mathbf{x}; \xi)] = \nabla f(\mathbf{x})$, and $\|\nabla F(\mathbf{x}; \xi) - \nabla f(\mathbf{x})\| \leq \sigma$ almost surely.*

     *(iv) The distribution of $\nabla F(\mathbf{x}; \xi)$ is symmetric around its mean $\nabla f(\mathbf{x})$, and the density is monotonically decreasing over the $\ell_2$ distance between the mean and the value of random variable.*

**Remark:**   The Assumption 1 (i) means that the loss function satisfies the relaxed-smoothness condition, and it holds when training a language model with LSTMs. We consider the homogeneous distributed learning setting, where each machine has access to same data distribution as in [44]. Assumption 1 (ii) and (iii) are standard assumptions in nonconvex optimization [9, 51]. Note that it is usually assumed that the stochastic gradient is unbiased and has bounded variance [9], but in the relaxed smoothness setting, we follow [51] to assume we have unbiased stochastic gradient with almost surely bounded deviation $\sigma$. Assumption 1 (iv) assumes the noise is unimodal and symmetric around its mean, which we empirically verify in Appendix E. Examples satisfying (iii) and (iv) include truncated Gaussian distribution, truncated student's t-distribution, etc.

---

**Algorithm 1** Communication Efficient Local Gradient Clipping (CELGC)

---

1: **for** $t = 0, \ldots, T$ **do**
2:     Each node $i$ samples its stochastic gradient $\nabla F(\mathbf{x}_t^i; \xi_t^i)$, where $\xi_t^i \sim \mathcal{D}$.
3:     Each node $i$ updates it local solution **in parallel**:

$$\mathbf{x}_{t+1}^i = \mathbf{x}_t^i - \min\left(\eta, \frac{\gamma}{\|\nabla F(\mathbf{x}_t^i; \xi_t^i)\|}\right)\nabla F(\mathbf{x}_t^i; \xi_t^i) \tag{2}$$

4:     **if** $t$ is a multiple of $I$ **then**
5:         Each worker resets the local solution as the averaged solution across nodes:

$$\mathbf{x}_t^i = \widehat{\mathbf{x}} := \frac{1}{N}\sum_{j=1}^{N}\mathbf{x}_t^j \qquad \forall i \in \{1, \ldots, N\} \tag{3}$$

6:     **end if**
7: **end for**

---

## 4 Algorithm and Theoretical Analysis

### 4.1 Main Difficulty and the Algorithm Design

We briefly present the main difficulty in extending the single machine setting [50] to the distributed setting. In [50], they split the contribution of decreasing objective value by considering two cases: clipping large gradients and keeping small gradients. If communication is allowed at every iteration, then we can aggregate gradients on each machine and determine whether we should clip or keep the averaged gradient or not. However, in our setting, communicating with other machines at every iteration is not allowed. This would lead to the following difficulties: (i) the averaged gradient may not be available to the algorithm if communication is limited, so it is hard to determine whether clipping operation should be performed or not; (ii) the model weight on every machine may not be the same when communication does not happen at the current iteration; (iii) the loss function is not $L$-smooth, so the usual local SGD analysis for $L$-smooth functions cannot be applied in this case.

To address this issue, we design a new algorithm, namely Communication-Efficient Local Gradient Clipping (CELGC), which is presented in Algorithm 1. The algorithm calculates a stochastic gradient and then performs multiple local gradient clipping steps on each machine in parallel, and aggregates model parameters on all machines after every $I$ steps of local updates. We aim to establish iteration and communication complexity for Algorithm 1 for finding an $\epsilon$-stationary point when $I > 1$.

### 4.2 A Lemma for Truncated Random Variables

As indicated in the previous subsection, Algorithm 1 clips gradient on each local machine. This feature greatly complicates the analysis: it is difficult to get an unbiased estimate of the stochastic gradient when its magnitude is not so large such that clipping does not happen. The reason is due to the dependency between random variables (i.e., stochastic gradient and the indicator of clipping). To get around of this difficulty, we introduce the following Lemma for estimating truncated random variables.

**Lemma 1.** *Denote by $\mathbf{g} \in \mathbb{R}^d$ a random vector. Suppose the distribution of $\mathbf{g}$ is symmetric around its mean, and the density is monotonically decreasing over the $\ell_2$ distance between the mean and the value of random variable, then there exists $\Lambda = diag(c_1, \ldots, c_d)$ and $0 < c_i \leq 1$ with $i = 1, \ldots, d$ such that*

$$\mathbb{E}\left[\mathbf{g}\mathbb{I}(\|\mathbf{g}\| \leq \alpha)\right] = Pr(\|\mathbf{g}\| \leq \alpha)\Lambda\mathbb{E}\left[\mathbf{g}\right], \tag{4}$$

*where $\alpha > 0$ is a constant, $\mathbb{I}(\cdot)$ is the indicator function.*

The proof of Lemma 1 is included in Appendix B. This Lemma provides an unbiased estimate for a truncated random variable. In the subsequent of this paper, we regard $\mathbf{g}$ in the Lemma as the stochastic gradient and $\alpha$ as the clipping threshold. In addition, we define $c_{\min} = \min(c_1, \ldots, c_d)$, and $c_{\max} = \max(c_1, \ldots, c_d)$. We have $0 < c_{\min} \leq c_{\max} \leq 1$.

## 4.3 Main Results

**Theorem 1.** *Suppose Assumption 1 holds and $\sigma \geq 1$. Take $\epsilon \leq \min(\frac{AL_0}{BL_1}, 0.1)$ be a small enough constant and $N \leq \min(\frac{1}{\epsilon}, \frac{14AL_0}{5BL_1\epsilon})$. In Algorithm 1, choose $I \leq \sqrt{\frac{1}{c_{\min}}} \frac{\sigma}{N\epsilon}$, $\gamma \leq c_{\min} \frac{N\epsilon}{28\sigma} \min\{\frac{\epsilon}{AL_0}, \frac{1}{BL_1}\}$ and the fixed ratio $\frac{\gamma}{\eta} = 5\sigma$, where $A \geq 1$ and $B \geq 1$ are constants which will be specified in the proof, and run Algorithm 1 for $T = O\left(\frac{\Delta L_0 \sigma^2}{N\epsilon^4}\right)$ iterations. Define $\bar{\mathbf{x}}_t = \frac{1}{N}\sum_{i=1}^{N} \mathbf{x}_t^i$. Then we have $\frac{1}{T}\sum_{t=1}^{T} \mathbb{E}\|\nabla f(\bar{\mathbf{x}}_t)\| \leq 9\epsilon$.*

**Remark:** We have some implications of Theorem 1. When the number of machines is not large (i.e., $N \leq O(1/\epsilon)$) and the number of skipped communications is not large (i.e., $I \leq O(\sigma/\epsilon N)$), then with proper setting of the learning rate, we have following observations. First, our algorithm enjoys linear speedup, since the number of iterations we need to find an $\epsilon$-stationary point is divided by the number of machines $N$ when comparing the single machine algorithm in [50]. Second, our algorithm is communication-efficient, since the communication complexity (a.k.a., number of rounds) is $T/I = O\left(\Delta L_0 \sigma \epsilon^{-3}\right)$, which provably improves the naive parallel gradient clipping algorithm of [50] with $O(\Delta L_0 \sigma^2/(N\epsilon^4))$ communication complexity when $N \leq O(1/\epsilon)$.

Another important fact is that both iteration complexity and communication complexity only depend on $L_0$ while being independent of $L_1$ and the gradient upper bound $M$. This indicates that our algorithm does not suffer from slow convergence even if these quantities are large. This is in line with [50] as well. In addition, local gradient clipping is a good mechanism to alleviate the bad effects brought by a rapidly changing loss landscape (e.g., some language models such as LSTM).

## 4.4 Sketch of the Proof of Theorem 1

In this section, we present the sketch of our proof of Theorem 1. The detailed proof can be found in Appendix C. The key idea in our proof is to establish the descent property of the sequence $\{f(\bar{\mathbf{x}}_t)\}_{t=0}^{T}$ in the distributed setting under the relaxed smoothness condition, where $\bar{\mathbf{x}}_t = \frac{1}{N}\sum_{i=1}^{t} \mathbf{x}_t^i$ is the averaged weight across all machines at $t$-th iteration. The main challenge is that the descent property of $(L_0, L_1)$-smooth function in the distributed setting does not naturally hold, which is in sharp contrast to the usual local SGD proof for $L$-smooth functions. To address this challenge, we need to carefully study whether the algorithm is able to decrease the objective function in different situations. Our main technical innovations in the proof are listed as the following.

First, we monitor the algorithm's progress in decreasing the objective value according to some novel measures. The measures we use are the magnitude of the gradient evaluated at the averaged weight and the magnitude of local gradients evaluated at the individual weights on every machine. To this end, we introduce Lemma 3, whose goal is to carefully inspect how much progress the algorithm makes, according to the magnitude of local gradients calculated on each machine. The reason is that the local gradient's magnitude is an indicator of whether the clipping operation happens or not. For each fixed iteration $t$, we define $J(t) = \{i \in [N] : \|\nabla F(\mathbf{x}_t^i, \xi_t^i)\| \geq \gamma/\eta\}$ and $\bar{J}(t) = [N] \setminus J(t)$. Briefly speaking, $J(t)$ contains all machines that perform clipping operation at iteration $t$ and $\bar{J}(t)$ is the set of machines that do not perform clip operation at iteration $t$. In Lemma 3, we perform the one-step analysis and consider all machines with different clipping behaviors at the iteration $t$. The proof of Lemma 3 crucially relies on the lemma for estimating truncated random variables (i.e., Lemma 1) to calculate the expectation of non-clipped gradients.

Second, Zhang et al. [50] inspect their algorithm's progress by considering the magnitude of the gradient at different iterations, so they treat every iteration differently. However, this approach does not work in the distributed setting with infrequent communication since one cannot get access to the averaged gradient across machines at every iteration. Instead, we treat every iteration of the algorithm as the same but consider the progress made by each machine.

Third, by properly choosing hyperparameters $(\eta, \gamma, I)$ and using an amortized analysis, we prove that our algorithm can decrease the objective value by a sufficient amount, and the sufficient decrease is mainly due to the case where the gradient is not too large (i.e., clipping operations do not happen). This important insight allows us to better characterize the training dynamics.

Now we present how to proceed with the proof in detail.

Lemma 2 characterizes the $\ell_2$ error between averaged weight and individual weights at $t$-th iteration.

**Lemma 2.** *Under Assumption 1, for any $i$ and $t$, Algorithm 1 ensures $\|\bar{\mathbf{x}}_t - \mathbf{x}_t^i\| \leq 2\gamma I$ holds almost surely.*

Lemma 5 and Lemma 6 (in Appendix A) are properties of $(L_0, L_1)$-smooth functions we need to use. To make sure they work, we need $2\gamma I \leq c/L_1$ for some $c > 0$. This is proved in the proof of Theorem 1 (in Appendix C).

Let $J(t)$ be the index set of $i$ such that $\|\nabla F(\mathbf{x}_t^i, \xi_t^i)\| \geq \frac{\gamma}{\eta}$ at fixed iteration $t$, i.e., $J(t) = \{i \in [N] \mid \|\nabla F(\mathbf{x}_t; \xi_t^i)\| \geq \gamma/\eta\}$. Lemma 3 characterizes how much progress we can get in one iteration of Algorithm 1, which is decomposed into contributions from every machine (note that $J(t) \cup \bar{J}(t) = \{1, \ldots, N\}$ for every $t$).

**Lemma 3.** *Let $J(t)$ be the set defined as above. If $2\gamma I \leq c/L_1$ for some $c > 0$, $AL_0\eta \leq 1/2$, and $\gamma/\eta = 5\sigma$, then we have*

$$\mathbb{E}[f(\bar{\mathbf{x}}_{t+1}) - f(\bar{\mathbf{x}}_t)]$$

$$\leq \frac{1}{N}\mathbb{E}\sum_{i \in J(t)}\left[-\frac{2\gamma}{5}\|\nabla f(\bar{\mathbf{x}}_t)\| - \frac{3\gamma^2}{5\eta} + \frac{50AL_0\eta^2\sigma^2}{N} + \frac{7\gamma}{5}\|\nabla F(\mathbf{x}_t^i; \xi_t^i) - \nabla f(\bar{\mathbf{x}}_t)\| + AL_0\gamma^2 + \frac{BL_1\gamma^2\|\nabla f(\bar{\mathbf{x}}_t)\|}{2}\right]$$

$$+ \mathbb{E}\frac{1}{N}\sum_{i \in \bar{J}(t)}\left[-\frac{\eta c_{\min}}{2}\|\nabla f(\bar{\mathbf{x}}_t)\|^2 + 4\gamma^2 I^2 A^2 L_0^2 \eta + 4\gamma^2 I^2 B^2 L_1^2 \eta\|\nabla f(\bar{\mathbf{x}}_t)\|^2 + \frac{50AL_0\eta^2\sigma^2}{N} + \frac{BL_1\gamma^2\|\nabla f(\bar{\mathbf{x}}_t)\|}{2}\right],$$

*where $A = 1 + e^c - \frac{e^c - 1}{c}$ and $B = \frac{e^c - 1}{c}$.*

Lemma 4 quantifies an upper bound of the averaged $\ell_2$ error between the local gradient evaluated at the local weight and the gradient evaluated at the averaged weight.

**Lemma 4.** *Suppose Assumption 1 holds. When $2\gamma I \leq c/L_1$ for some $c > 0$, the following inequality holds for every $i$ almost surely with $A = 1 + e^c - \frac{e^c - 1}{c}$ and $B = \frac{e^c - 1}{c}$:*

$$\left\|\nabla F(\mathbf{x}_t^i; \xi_t^i) - \nabla f(\bar{\mathbf{x}}_t)\right\| \leq \sigma + 2\gamma I(AL_0 + BL_1\|\nabla f(\bar{\mathbf{x}}_t)\|).$$

**Putting all together** Suppose our algorithm runs $T$ iterations. Taking summation on both sides of Lemma 3 over all $t = 0, \ldots, T - 1$, we are able to get an upper bound of $\sum_{t=0}^{T-1} \mathbb{E}[f(\bar{\mathbf{x}}_{t+1}) - f(\bar{\mathbf{x}}_t)] = \mathbb{E}[f(\bar{\mathbf{x}}_T) - f(\bar{\mathbf{x}}_0)]$. Note that $\mathbb{E}[f(\bar{\mathbf{x}}_T) - f(\bar{\mathbf{x}}_0)] \geq -\Delta$ due to Assumption 1, so we are able to get an upper bound of gradient norm. For details, please refer to the proof of Theorem 1 in Appendix C.

## 5 Experiments

We conduct extensive experiments to validate the merits of our algorithm on various tasks (e.g., image classification, language modeling). In the main text below, we consider the homogeneous data setting where each machine has the same data distribution and every machine participate the communication at each round. In the Appendix D, we also test our algorithm in the general federated learning setting (e.g., partial participation of machines), ablation study on small/large batch-sizes, and other experiments, etc.

We conduct each experiment in two nodes with 4 Nvidia-V100 GPUs on each node. In our experiments, one "machine" corresponds to one GPU, and we use the word "GPU" and "machine" in this section interchangeably. We compared our algorithm with the baseline across three deep learning benchmarks: CIFAR-10 image classification with ResNet, Penn Treebank language modeling with LSTM, and Wikitext-2 language modeling with LSTM, and ImageNet classification with ResNet. All algorithms and the training framework are implemented in Pytorch 1.4. Due to limited computational resources, for our algorithms, we choose same hyperparameters like the clipping thresholds according to the best-tuned baselines unless otherwise specified.

We compare our algorithm of different $I$ with the baseline, which is the naive parallel version of the algorithm in [50]. We want to re-emphasize that the difference is that the baseline algorithm needs to average the model weights and local gradients at every iteration while ours only requires averaging the model weights after every $I$ iterations. We find that our algorithm with a range of values of $I$ can

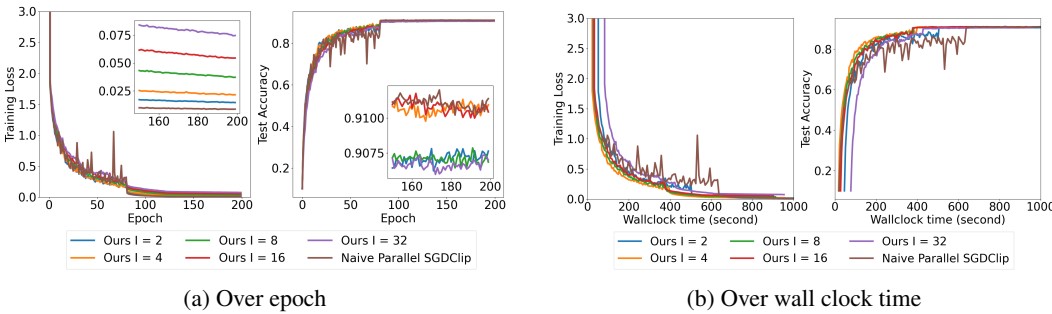

(a) Over epoch

(b) Over wall clock time

Figure 1: Algorithm 1 with different $I$: Training loss and test accuracy v.s. (Left) epoch and (right) wall clock time on training a 56 layer Resnet to do image classification on CIFAR10.

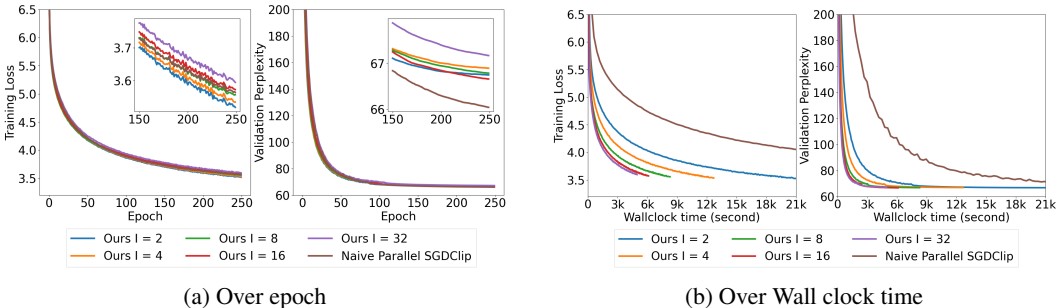

(a) Over epoch

(b) Over Wall clock time

Figure 2: Algorithm 1 with different $I$: Training loss and validation perplexity v.s. (Left) epoch and (right) wall clock time on training an AWD-LSTM to do language modeling on Penn Treebank.

match the results of the baseline in terms of epochs on different models and data. This immediately suggests that our algorithm will gain substantial speedup in terms of the wall clock time, which is also supported by our experiments.

## 5.1 Effects of Skipping Communication

We focus on one feature of our algorithm: skipping communication rounds. Theorem 1 says that our algorithm enjoys reduced communication complexity since every node only communicates with other nodes periodically with node synchronization interval length $I$. To study how communication skipping affects the convergence of Algorithm 1, we run it with $I \in \{2, 4, 8, 16, 32\}$.

**CIFAR-10 classification with ResNet-56.** We train the standard 56-layer ResNet [15] architecture on CIFAR-10. We use SGD with clipping as the baseline algorithm with a stagewise decaying learning rate schedule, following the widely adopted fashion on training the ResNet architecture. Specifically, we use the initial learning rate $\eta = 0.3$, the clipping threshold $\gamma = 1.0$, and decrease the learning rate by a factor of 10 at epoch 80 and 120. The local batch size at each GPU is 64. These parameter settings follow that of [47].

The results are illustrated in Figure 1. Figure 1a shows the convergence of training loss and test accuracy v.s. the number of epochs that are jointly accessed by all GPUs. This means that, if the x-axis value is 8, then each GPU runs 1 epoch of training data. The same convention applied to all other figures for multiple GPU training in this paper. Figure 1b verifies our algorithm's advantage of skipping communication by plotting the convergence of training loss and test accuracy v.s. the wall clock time. Overall, we can clearly see that our algorithm matches the baseline epoch-wise but greatly speeds up wall-clock-wise.

**Language modeling with LSTM on Penn Treebank.** We adopt the 3-layer AWD-LSTM [31] to do language modeling on Penn Treebank (PTB) dataset [27] (word level). We use SGD with clipping as the baseline algorithm with the clipping threshold $\gamma = 7.5$. The local batch size at each GPU is 3. These parameter settings follow that of [50]. We fine-tuned the initial learning rate $\eta$ for all algorithms (including the baseline) by choosing the one giving the smallest final training loss in the range $\{0.1, 0.5, 1, 5, 10, 20, 30, 40, 50, 100\}$.

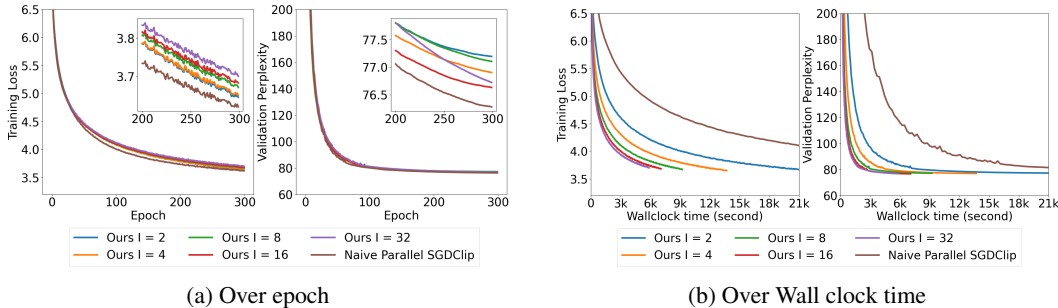

(a) Over epoch                              (b) Over Wall clock time

Figure 3: Algorithm 1 with different $I$: Training loss and validation perplexity v.s. (Left) epoch and (right) wall clock time on training an AWD-LSTM to do language modeling on Wikitext-2.

We report the results in Figure 2. It can be seen that we can match the baseline in both training loss and validation perplexity epoch-wise while gaining substantial speedup (4x faster for $I = 4$) wall-clock-wise.

**Language modeling with LSTM on Wikitext-2.** We adopt the 3-layer AWD-LSTM [31] to do language modeling on Wikitext-2 dataset [27](word level). We use SGD with clipping as the baseline algorithm with the clipping threshold $\gamma = 7.5$. The local batch size at each GPU is 10. These parameter settings follow that of [31]. We fine-tuned the initial learning rate $\eta$ for all algorithms (including the baseline) by choosing the one giving the smallest final training loss in the range $\{0.1, 0.5, 1, 5, 10, 20, 30, 40, 50, 100\}$.

We report the results in Figure 3. We can match the baseline in both training loss and validation perplexity epoch-wise, but we again obtain large speedup (4x faster for $I = 4$) wall-clock-wise. This, together with the above two experiments, clearly show our algorithm's effectiveness in speeding up the training in distributed settings. Another observation is that Algorithm 1 can allow relatively large $I$ without hurting the convergence behavior.

**ImageNet Classification with ResNet-50** We compared our algorithm with several baselines in training a ResNet-50 on ImageNet. We compared with two strong baselines: one is the Naive Parallel SGDClip, another is a well-accepted baseline for ImageNet by [12]. We run the experiments on 8 GPUs. We follow the settings of [12] to setup hyperparameter for baselines. Specifically, for every method, the initial learning rate is 0.0125, and we use the warmup with 5 epochs, batch size 32, momentum parameter 0.9, the weight decay $5 \times 10^{-4}$. The learning rate multiplying factor is 1 for the epoch $5 \sim 30$, and 0.1 for epoch $30 \sim 60$, 0.01 for epochs $60 \sim 80$, and 0.001 for epoch $80 \sim 90$. The clipping threshold for Naive Parallel SGDClip and our method CELGC are both set to be 1. We consider our algorithm with $I = 4$, i.e., our algorithm CELGC performs weight averaging after 4 steps of local stochastic gradient descent with gradient clipping on each GPU.

The results are shown in Figure 6. We report the performance of these methods from several different perspectives (training accuracy/validation accuracy versus epoch, and training accuracy/validation accuracy versus wallclock time). We can see that the training accuracy of our algorithm CELGC with $I = 4$ can match both baselines in terms of epoch (Figure 6a), but it is much better in terms of running time (Figure 6b).

## 5.2 Verifying Parallel Speedup

Figure 4 show the training loss and test accuracy v.s. the number of iterations. In the distributed setting, one iteration means running one step of Algorithm 1 on all machines; while in the single machine setting, one iteration means running one step of SGD with clipping. In our experiment, we use minibatch size 64 on every GPU in the distributed setting to run Algorithm 1, while we also use 64 minibatch size on the single GPU to run SGD with clipping. In the left two panels of Figure 4, we can clearly find that even with $I > 1$, our algorithm still enjoys parallel speedup, since our algorithm requires less number of iterations to converge to the same targets (e.g., training loss, test accuracy). This observation is consistent with our iteration complexity results in Theorem 1.

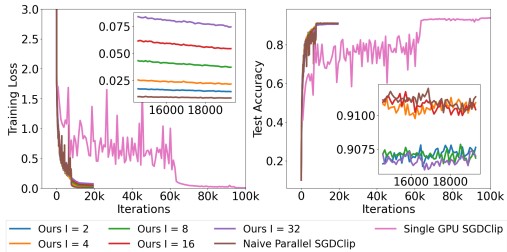
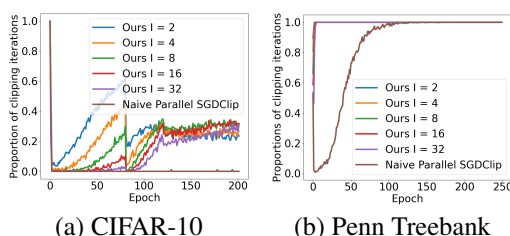

(a) CIFAR-10      (b) Penn Treebank

Figure 4: Performance v.s. # of iterations each GPU runs on training ResNet-56 on CIFAR-10 showing the parallel speedup.

Figure 5: Proportions of iterations in each epoch in which clipping is triggered v.s. epochs showing clipping is very frequent.

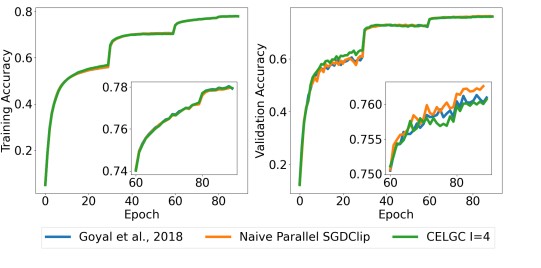
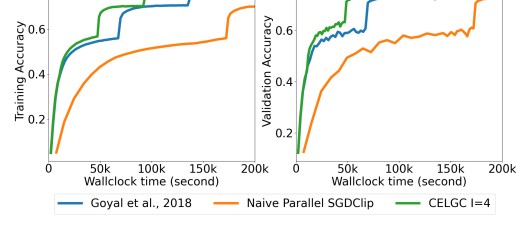

(a) Performance over epoch        (b) Performance over wall clock time

Figure 6: Training loss and test accuracy v.s. epoch (left) and wall clock time (right) on training a Resnet-50 to do image classification on ImageNet.

## 5.3 Clipping Operation Happens Frequently

Figure 5 reports the proportion of iterations in each epoch that clipping is triggered. We observe that for our algorithm, clipping happens more frequently than the baseline, especially for NLP tasks. We conjecture that this is because we only used local gradients in each GPU to do the clipping without averaging them across all machines as the baseline did. This leads to more stochasticity of the norm of the gradient in our algorithm than the baseline, and thus causes more clippings to happen. This observation highlights the importance of studying clipping algorithms in the distributed setting. Another interesting observation is that clipping happens much more frequently when training language models than image classification models. Hence this algorithm is presumably more effective in training deep models in NLP tasks.

## 6 Conclusion

In this paper, we design a communication-efficient distributed stochastic local gradient clipping algorithm to train deep neural networks. By exploring the relaxed smoothness condition which was shown to be satisfied for certain neural networks, we theoretically prove both the linear speedup property and the improved communication complexity when the data distribution across machines is homogeneous. Our empirical studies show that our algorithm indeed enjoys parallel speedup and greatly improves the runtime performance in various federated learning scenarios (e.g., partial client participation). One limitation of our work is that our convergence analysis is only applicable for homogeneous data, and it would be interesting to analyze the settings of heterogeneous data theoretically in the future.

## Acknowledgements

We would like to thank the anonymous reviewers for their help comments. Mingrui Liu is supported by a grant at George Mason University. Computations were run on ARGO, a research computing cluster provided by the Office of Research Computing at George Mason University (URL: https://orc.gmu.edu). The majority of work of Zhenxun Zhuang was done when he was a Ph.D. student at Boston University.

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
