# OpenReview forum: "A Communication-Efficient Distributed Gradient Clipping Algorithm for Training Deep Neural Networks"
_NeurIPS.cc/2022/Conference — NeurIPS 2022 Accept_

### Official Review · Reviewer_ucfd · 2022-06-26

**Rating:** 5
**Confidence:** 4
**Soundness:** 2 fair
**Presentation:** 2 fair
**Contribution:** 3 good

**Summary:**

The paper focuses on training neural networks in the infrequent-communication setting (such as Local-SGD) with gradient clipping. Prior works have shown that gradient clipping is an important technique to stabilize the training of neural networks. Local-SGD is a communication-efficient distributed training algorithm that can significantly reduce the communication overhead and lead to totraining speedup. However, using the two is not straightforward as gradient clipping is applied on the averaged gradient whereas Local-SGD conducts several local steps before executing any communications.

To overcome this issue the paper proposes CELGC, which relies on prior work by [1]. The paper provides a theoretical and empirical study of CELGC. The experiments include an evaluation of the ResNet-34 and AWD-LSTM neural architectures.


[1] Zhang, Bohang, et al. "Improved analysis of clipping algorithms for non-convex optimization." Advances in Neural Information Processing Systems 33 (2020): 15511-15521.

**Questions:**

1. It is common today to train neural networks with more advanced optimizers ( such as Momentum or Adam) than vanilla SGD which was used in the paper. What is the compatibility of CELGC with these relatively newer optimizers?
2. Today, it is common to apply gradient clipping locally in the Local-SGD or federated setting. Why was CELGC not compared to this simple approach?
3. The Naive Parallel SGDClip was defined as using [3] and synchronizing at every iteration. Why can't [3] simply be applied locally when training in the Local-SGD setting?
4. In Figure 1 the Naive Parallel SGDClip is much more volatile compared to CELGC, why is that?

[1] Kingma, Diederik P., and Jimmy Ba. "Adam: A method for stochastic optimization." arXiv preprint arXiv:1412.6980 (2014).

[2] Zhang, Xinwei, et al. "Understanding Clipping for Federated Learning: Convergence and Client-Level Differential Privacy." arXiv preprint arXiv:2106.13673 (2021).

[3] Zhang, Bohang, et al. "Improved analysis of clipping algorithms for non-convex optimization." Advances in Neural Information Processing Systems 33 (2020): 15511-15521.

---
> Improving the empirical study would help improve the score:
- Stronger empirical results, such as ResNet-50 on ImagNet.
- Comparison to the "naive" algorithms stated above (2 and 3).
- Adding zoomed-in figures to see the convergence rate.
- Listing the final results in a table.
- Hyperparameter robustness tests.
- Confidence intervals for figures.


**Limitations:**

The paper addresses some of the limitations of CELGC in the conclusion section. For example, the limitation of being restricted to homogenous data makes it incompatible with the federated setting where the data is often heterogeneous.  A second limitation, which is not discussed, is that the CELGC requires tuning \gamma as it cannot be directly inferred from the existing gradient clipping hyperparameter. This adds additional computational cost when using CELGC on an existing neural architecture.

**Strengths And Weaknesses:**

> **TL;DR:**
The paper presents an algorithm called CELGC that makes gradient clipping compatible with the Local-SGD setting. The paper provides both a theoretical and an empirical study of the algorithm. However, the algorithm is lacking novelty due to the high similarity with [1] and completely disregards very relevant papers such as [2]. Furthermore, the empirical results are hard to read and not convincing enough.
---
**Update Following Rebuttal with imageNet Results**
Throughout the rebuttal process, the authors have responded to my concerns and I appreciate the answers. The authors have made great efforts to improve the paper by adding confidence intervals to the figures and clearly listing the final accuracies. The authors also added ImageNet experiments that both show strong empirical results and the usefulness of the proposed CELGC algorithm. I'm still concerned about usefulness of the CELGC, since it requires re-tuning the $\gamma$ hyper-parameter. However, I believe that CELGC is robust and provides both strong empirical and theoretical results. Therefore, I think that this paper should accepted after its rebuttal improvements.
---

**Pros:**
1. The paper is well written and addresses an important problem that allows Local-SGD to use gradient clipping.
2. The theoretical analysis seems correct with no significant errors.
3. The experiments include many different configurations, such as a federated learning experiment with homogenous data (although heterogeneous data would be preferable in this configuration).
4. The experiments used a learning-rate hyperparameter grid search for tuning each algorithm.

**Cons:**
1. The Figures in the experiment section are hard to read and often confusing. For example, in the two left plots in Figure 4 the comparison of convergence speed or final accuracy to the single GPU (baseline) is not possible. This specific figure is also misleading, as the X-axis is not the amount of processed data but rather some sort of execution time combined with iterations. Adding a zoomed-in figure and listing the final results in a table would significantly improve this. Furthermore, many figures are missing a valid comparison to the single GPU scenario (baseline) in order to show true linear scalability.
2. The figures do not have error bars or any confidence intervals. This is crucial for justifying the empirical results. Furthermore, I expect a robustness evaluation as listing the best-tuned results does not provide the whole picture.
3. The two right plots in Figure 4 are not described and can only be understood after collecting the pieces of information myself and solving the puzzle. It is not clearly stated which plot refers to which neural network. Figure 4 should be split into two separate plots (the two right and two left) with proper sub-captions.
4. A comparison to known scalable architecture-dataset combination is required. Figure 10 does not convince the scalability as it is not evaluated on a scalable configuration, such as ResNet-50 on Imagenet that was used in [1].

**Typos:**
1. Line #32: "in federated" --> "in the federated"
2. Algorithm 1 Line #3: "it" --> "its"
3. Line #239: "the iteration" --> "iteration"

[1] Zhang, Bohang, et al. "Improved analysis of clipping algorithms for non-convex optimization." Advances in Neural Information Processing Systems 33 (2020): 15511-15521.

[2] Zhang, Xinwei, et al. "Understanding Clipping for Federated Learning: Convergence and Client-Level Differential Privacy." arXiv preprint arXiv:2106.13673 (2021).

---

> ### Author Response · Authors · 2022-07-29
> **Thank you for your review. Please check out our responses [2/2]**
>
> **5. A comparison to known scalable architecture-dataset combination is required. Figure 10 does not convince the scalability as it is not evaluated on a scalable configuration, such as ResNet-50 on Imagenet that was used in [1].**
>
> > A: Please note that our main focus in this paper is to study the training of deep neural networks when the clipping step is indispensable such as in training language models. As we wrote in Section 5.3, clipping happens much more frequently when training language models than image classification models. Thus, we devote the majority of our efforts to NLP tasks. We are running experiments on training ResNet-50 on Imagenet right now and expect to include it in the final version.
>
> **6. It is common today to train neural networks with more advanced optimizers ( such as Momentum or Adam) than vanilla SGD which was used in the paper. What is the compatibility of CELGC with these relatively newer optimizers?**
>
> > A: We have run an experiment comparing our algorithm with naive Parallel SGDClip when both are equipped with momentum. Results are reported in Section D.11 showing that our algorithm couples well with momentum.
>
> **7. Today, it is common to apply gradient clipping locally in the Local-SGD or federated setting. Why was CELGC not compared to this simple approach? The Naive Parallel SGDClip was defined as using [3] and synchronizing at every iteration. Why can't [3] simply be applied locally when training in the Local-SGD setting?**
>
> > A: We believe that CELGC is indeed the “simple” approach of local SGD with local gradient clipping. Could you please give us more details on the method we need to compare? We can include it in the revised version.
>
> **8. In Figure 1 the Naive Parallel SGDClip is much more volatile compared to CELGC, why is that?**
>
> > A: It is hard to identify the actual reason for your concern. Yet, we noticed from Figure 5(a) that the Naive Parallel SGDClip performs very few clipping steps during the entire training phase, whereas CELGC performs much more clipping steps. We conjecture that this could be a possible reason as an un-clipped stochastic gradient could be much more noisy than clipped ones.
>
> **9. For example, the limitation of being restricted to homogenous data makes it incompatible with the federated setting where the data is often heterogeneous. A second limitation, which is not discussed, is that the CELGC requires tuning $\gamma$ as it cannot be directly inferred from the existing gradient clipping hyperparameter. This adds additional computational cost when using CELGC on an existing neural architecture.**
>
> > A: For the first limitation, though we did not show theoretical results for the heterogeneous setting, we have conducted an experiment on such a setting showing that CELGC can still match the Naive Parallel SGDClip. Please see Section D.9 in the updated version.
> >
> > For the second limitation, we actually inherited the clipping parameter $\gamma$ from previous works. Specifically, $\gamma = 1$ for CIFAR10 and $\gamma = 7.5$ for Penn Treebank both follow [Zhang et al., NeurIPS 2020].
> >
>
> > [Zhang et al., NeurIPS 2020] Bohang Zhang, Jikai Jin, Cong Fang, and Liwei Wang. Improved analysis of clipping algorithms for non-convex optimization. In Advances in Neural Information Processing Systems. 2020.

---

> > ### Comment · Reviewer_ucfd · 2022-08-05
> > **Follow-up #2**
> >
> > 5. There are many NLP scalable configuration that are applicable here, such as BERT. See MLPerf for other options: https://mlcommons.org/en/
> >
> > 6. The referred results in the appendix section D.11 seem to be newly added results. This results are in the right direction but are missing error bars. Furthermore, there is no justification to how the momentum coefficient was chosen.
> >
> > 7. The simple baseline I referred is to apply gradient clipping locally when training with Local-SGD.
> >
> > 8. The Naive Parallel SGDClip in essence mimics training with one large batch and best reflects the "true" training configuration. This might point that the chosen hyper-parameters were not correct and strengthen my claims in (5) that these paper lacks strong scalable baselines.
> >
> > 9. CELGC still requires tuning $\gamma$ as it cannot be directly inferred from the existing gradient clipping hyper-parameter.

---

> > > ### Author Response · Authors · 2022-08-05
> > > **Thank you for your follow-up and we address your concerns below**
> > >
> > > 5. **There are many NLP scalable configuration that are applicable here, such as BERT. See MLPerf for other options: https://mlcommons.org/en/**
> > >
> > > > A: Please note that we are not sure whether BERT satisfies our assumption (i.e., $(L_0,L_1)$-smoothness), so we did not try BERT in our current version.
> > > In addition, we are running imagenet experiments as you requested, so it might be difficult for us to run BERT simultaneously with ImagetNet in such a short author response period. We will leave it as future work.
> > >
> > > 6. **The referred results in the appendix section D.11 seem to be newly added results. This results are in the right direction but are missing error bars. Furthermore, there is no justification to how the momentum coefficient was chosen.**
> > >
> > > > A: Please note that we already added lots of experiments during the one week response period, so we mainly focused on our original algorithm without momentum for which the multiple run results are added in Section D.12. It is easy to see that the performance is pretty stable and we believe the same phenomenon would also hold for momentum version of our algorithm. We are going to add the error bar experiments for the momentum variant in the revised version.
> > > >
> > > > Also, as we wrote, we inherited the momentum coefficient from [Zhang et al., NeurIPS 2020] as $\beta = 0.9$ (from Figure 2(c) in their paper). We believe $\beta=0.9$ is a common practice in deep learning and it does not need to be tuned.
> > > >
> > > > [Zhang et al., NeurIPS 2020] Bohang Zhang, Jikai Jin, Cong Fang, and Liwei Wang. "Improved analysis of clipping algorithms for non-convex optimization." Advances in Neural Information Processing Systems 33 (2020): 15511-15521.
> > >
> > > 7. **The simple baseline I referred is to apply gradient clipping locally when training with Local-SGD.**
> > >
> > > > A: We are not sure why our algorithm (CELGC) is different from local SGD with local gradient clipping. Could you please elaborate more why they are different?
> > >
> > > 8. **The Naive Parallel SGDClip in essence mimics training with one large batch and best reflects the "true" training configuration. This might point that the chosen hyper-parameters were not correct and strengthen my claims in (5) that these paper lacks strong scalable baselines.**
> > >
> > > > A: Please note that Naive Parallel SGDClip is mathematically equivalent to [Zhang et al. NeurIPS 2020] since each machine is processing $1/8$ batchsize data and we have $8$ machines, so the best parameter in [Zhang et al. NeurIPS 2020] is indeed the best parameter for Naive Parallel SGDClip.
> > > >
> > > > For large-scale experiment, as we said, we are running imagenet and expect to include it in the final version.
> > >
> > > 9. **CELGC still requires tuning as it cannot be directly inferred from the existing gradient clipping hyper-parameter.**
> > >
> > > > A: We respectfully disagree. Please note that our algorithm has good performance even if we simply inherited the hyperparmeters from existing gradient clipping algorithm. If we further tune these hyperparameters, we can get even better performance. We believe this is a strength rather than weakness for our algorithm.

---

> > > > ### Comment · Reviewer_ucfd · 2022-08-05
> > > > **Follow-up #3**
> > > >
> > > > 5. I understand that due to the short time BERT is out of the scope. However, I was referring to the claim that most efforts were towards NLP tasks, and in my opinion the current experiments lack strong benchmark results.
> > > >
> > > > 6. I accept your answer.
> > > >
> > > > 7. I understand, but it shows a lack of novelty by just combining two techniques that are already used today together in common practice. However, I do appreciate the theoretical work and therefore believe that it does help overcome this issue.
> > > >
> > > > 8. With that said, the high volatility of Naive Parallel SGDClip in Figure 1 is concerning.
> > > >
> > > > 9. Different training architectures require different gradient clipping coefficient. Since CELGC's $\gamma$ cannot be derived directly from the gradient clipping coefficient it would require re-tuning for good results.
> > > >
> > > > **Bottom Line**
> > > > I appreciate the author efforts and would recommend to maintain the the responses as I would like to be convinced. I currently have 3 top issues that would help me increase the score:
> > > > 1. Lack of strong empirical benchmarks -the promised ImageNet experiments should be sufficient given the results.
> > > >
> > > > 2. Re-tuning of $\gamma$ - I'm yet convinced that re-tuning is not required for each neural architecture that CELGC hasn't been trained on before.
> > > >
> > > > 3. Tables 3 show that with ResNet-56 on CIFAR-10 the final accuracy of CELGC can drop by up to 3% compared to the single worker baseline. This could not be seen in the previous figures and I appreciate the added clarity. However, due to such large drop in accuracy I do find CELGC less impactful. Showing strong empirical results on ImageNet with final accuracy and zoomed results can overcome this issue.

---

> > > > > ### Author Response · Authors · 2022-08-09
> > > > > **Thank you so much for your further feedback and we would like to address below.**
> > > > >
> > > > > Dear Reviewer ucfd,
> > > > >
> > > > > We would like to sincerely thank you for your valuable suggestions. We have conducted imagenet experiments and added results in Appendix D.14. Our algorithm has got very good performance: our algorithm spent only $44$ epochs and reaches a validation accuracy of $80\\%$, which is even better than a strong baseline trained with $90$ epochs [Goyal et al. 2018]: our algorithm is much better in terms of both epochs and wallclock time. Please check more details in the revised paper.
> > > > >
> > > > > Due to the short rebuttal time period, we would like to mention that the training for both NaiveParallel SGDClip and our algorithm CELGC is not finished yet (the training epochs are less than 90 as in [Goyal et al. 2018]). The training for the NaiveParallel SGDClip method is very slow due to expensive communication, and we also haven’t finished the training of our algorithm (we have only run $44$ epochs so far). We are still running them. However, according to the results we have, NaiveParallel SGDclip seems to be not practical in terms of wallclock time, and our algorithm CELGC is already better than [Goyal et al. 2018] even though it only runs $44$ epochs. We will let you know the results when the $90$ epochs training is finished.
> > > > >
> > > > > Given these promising results on ImageNet, as well as our newly added experiments in the rebuttal, we have tried our best and lots of efforts trying to improve the paper. We therefore kindly ask you to consider increasing the rating of our paper. We are happy to address your future concerns. Thanks!
> > > > >
> > > > > [Goyal et al. 2018] Goyal, P., Dollár, P., Girshick, R.B., Noordhuis, P., Wesolowski, L., Kyrola, A., Tulloch, A., Jia, Y., & He, K. (2017). Accurate, Large Minibatch SGD: Training ImageNet in 1 Hour. ArXiv, abs/1706.02677.
> > > > >
> > > > > Best,
> > > > > Authors

---

> > > > > > ### Comment · Reviewer_ucfd · 2022-08-09
> > > > > > **Follow-up #4**
> > > > > >
> > > > > > As I stated in the main post, the ImageNet results are indeed impressive. Please see my comments there.
> > > > > >
> > > > > > I've updated the score accordingly.

---

> ### Author Response · Authors · 2022-07-29
> **Thank you for your review. Please check out our responses [1/2]**
>
> **1. In the two left plots in Figure 4 the comparison of convergence speed or final accuracy to the single GPU (baseline) is not possible. This specific figure is also misleading, as the X-axis is not the amount of processed data but rather some sort of execution time combined with iterations. Adding a zoomed-in figure and listing the final results in a table would significantly improve this.**
>
> > A: The purpose of Figure 4 is to show that the number of iterations for our algorithm is reduced due to parallel speedup, compared with single GPU baseline. Indeed, the single GPU baseline processes 64 images every iteration, and our algorithm processes 64 images on every machine at every iteration (64*8 images in total with 8 machines per iteration). We have explained it at Section 5.2. We will make it more clear in the revised version.
> >
> > We have also added zoomed-in figures and summarized results in Tables 3, 4, and 5.
>
> **2. Many figures are missing a valid comparison to the single GPU scenario (baseline) in order to show true linear scalability.**
>
> > A: Actually, we devoted Section D.7 to verifying the linear speedup property of our algorithm.
>
> **3. The figures do not have error bars or any confidence intervals. Furthermore, I expect a robustness evaluation as listing the best-tuned results does not provide the whole picture.**
>
> > A: We have run the Penn Treebank experiment $3$ times with different random seeds. The results are reported in Section D.12 showing that our algorithm is stable.
>
> **4. The two right plots in Figure 4 are not described and can only be understood after collecting the pieces of information myself and solving the puzzle. It is not clearly stated which plot refers to which neural network. Figure 4 should be split into two separate plots (the two right and two left) with proper sub-captions.**
>
> > A: Sorry for the confusion. We have made the change (splitting Figure 4 as Figure 4 and Figure 5) accordingly. Please check our updated version. Thank you.

---

> > ### Comment · Reviewer_ucfd · 2022-08-04
> > **Follow-up #1**
> >
> > 1. A zoomed in figure would be hard to compare as the baseline and the parallel experiments are currently on a different scale. It would be much better to align the experiments on the same scale by setting the x-axis to the number epochs as commonly done in previous works. A table with final accuracies would also add some clarity.
> >
> > 2. Section D.7 is lacking many experiments and is not convincing.
> >
> > 3. Adding the error bars would add clarity.
> >
> > 4. The titles of the figures now describe the shown results.

---

> > > ### Author Response · Authors · 2022-08-05
> > > **Thank you for your follow-up and we address your concerns below**
> > >
> > > 1. **A zoomed in figure would be hard to compare as the baseline and the parallel experiments are currently on a different scale. It would be much better to align the experiments on the same scale by setting the x-axis to the number epochs as commonly done in previous works. A table with final accuracies would also add some clarity.**
> > >
> > > > A: The final results for each experiment are already reported in Table 2, 3, and 4 on Page 29 - 30 and we encourage the reviewer to check them out.
> > >
> > > 2. **Section D.7 is lacking many experiments and is not convincing.**
> > >
> > > > A: The experiment results we showed in Figure 12 clearly conveyed the message that we can match the single GPU SGDClip algorithm epoch-wise and thus achieve the linear speedup. Also, the final results for each experiment are already reported in Table 2, 3, and 4 on Page 29 - 30. Could you please elaborate more on why they are not convincing?
> > >
> > > 3. **Adding the error bars would add clarity.**
> > >
> > > > A: Regarding your concern, for the Penn Treebank experiments as an example given the time constraint, we have added the shading for each line showing the 95% confidence interval computed across 3 independent runs from different random seeds in Figure 17 which should serve similar purposes as error bars and we have also reported the results of multiple runs in Table 2.

---

> > > > ### Comment · Reviewer_ucfd · 2022-08-05
> > > > **See Follow-up #3**
> > > >
> > > > Thank you for the comments, please see bottom line statement in Follow-up #3.

---

### Official Review · Reviewer_xkbT · 2022-07-09

**Rating:** 4
**Confidence:** 4
**Soundness:** 2 fair
**Presentation:** 3 good
**Contribution:** 2 fair

**Summary:**

This paper studied the local stochastic gradient clipping method for nonconvex distributed optimization. Meanwhile, it provided the convergence analysis of the proposed CELGC algorithm under i.i.d. setting. It also conducted the empirical experiments to verify the efficiency of the proposed algorithm.

**Questions:**

1)	From algorithm’ view, the proposed CELGC algorithm only extend the classic FedAvg algorithm to  gradient clipping setting.

2)	From theoretical analysis’ view, although the given convergence analysis relax the smoothness condition, it strictly relies on the i.i.d. setting. In convergence analysis, the given relaxation of the smoothness condition is also popular, so it does not take any challenge.

3)	In the experiments, the given comparative methods only including gradient clipping methods are not reasonable. You should add some existing standard gradient-based methods such as FedAvg as comparative methods.

**Limitations:**

Yes

**Strengths And Weaknesses:**

This paper studied the local stochastic gradient clipping method for nonconvex distributed optimization. Meanwhile, it provided the convergence analysis of the proposed CELGC algorithm under i.i.d. setting. It also conducted the empirical experiments to verify the efficiency of the proposed algorithm.

1)	From algorithm’ view, the proposed CELGC algorithm only extend the classic FedAvg algorithm to  gradient clipping setting.

2)	From theoretical analysis’ view, although the given convergence analysis relax the smoothness condition, it strictly relies on the i.i.d. setting. In convergence analysis, the given relaxation of the smoothness condition is also popular, so it does not take any challenge.

Overall, the novelties and contributions of this paper do not reach the level of NeurIPS.

---

> ### Author Response · Authors · 2022-07-29
> **Thank you for your review. Please check our reponses below.**
>
>
> **1. From algorithm’ view, the proposed CELGC algorithm only extend the classic FedAvg algorithm to  gradient clipping setting.**
>
> > A: It is indeed a natural extension from FedAvg to gradient clipping. However, even this simple algorithm does not have any convergence analysis with linear speedup and reduced communication round. It is a highly nontrivial task and deserves a paper to study it.
>
> **2. From theoretical analysis’ view, although the given convergence analysis relax the smoothness condition, it strictly relies on the i.i.d. setting. In convergence analysis, the given relaxation of the smoothness condition is also popular, so it does not take any challenge.**
>
> > A: We respectfully disagree. The popularity of relaxed smoothness does not imply that it does not take any challenge. Even in the i.i.d. setting, the proof is nontrivial and significantly deviates from the previous analysis (e.g, [50, 51]). The task is indeed challenging (we highlight the challenges in Section 4.1). To show the linear speedup and reduced communication complexity, we need to introduce a new analysis roadmap (Section 4.4) and develop new tools for decoupling the dependency of random variables (Lemma 1). We believe this result is strong and significant.
>
>
> **3. In the experiments, the given comparative methods only including gradient clipping methods are not reasonable. You should add some existing standard gradient-based methods such as FedAvg as comparative methods.**
>
> >A: We indeed compared our algorithm with FedAvg in Figure 13 in Appendix Section D.8. It shows that gradient clipping gives significant advantages over FedAvg when training LSTMs on Penn Treebank dataset. We will make it more clear in the revised version.

---

> > ### Comment · Reviewer_xkbT · 2022-08-09
> > **Thanks for the authors' responses**
> >
> > Thanks for the authors'  responses.
> >
> > I have read all reviews and responses. I still concern the novelty of this paper:
> >
> > 1) From algorithm’ view, the proposed CELGC algorithm only extend the classic FedAvg algorithm to  gradient clipping setting.
> >
> > 2) From theoretical analysis’ view, although the given convergence analysis relax the smoothness condition, it strictly relies on the i.i.d. setting. In convergence analysis, the given relaxation of the smoothness condition is also popular, so it does not take any challenge.
> >
> > Thus, I still keep my score.

---

> > > ### Author Response · Authors · 2022-08-10
> > > **Please consider our response**
> > >
> > > Dear Reviewer xkbT,
> > >
> > > We kindly suggest you to update your evaluation model when you have more new information from us. It seems that you are still asking exactly the same questions even after reading our response. It would make us difficult to further clarify your concerns.
> > >
> > > Could you please let us know what exactly your concern is, after reading our rebuttal? We will try our best to help you better understand the technical contribution of our paper.
> > >
> > > Best,
> > > Authors

---

> ### Author Response · Authors · 2022-08-08
> **Look forward to post-rebuttal feedback!**
>
> Dear Reviewer xkbT,
>
> Thank you for your time reviewing our paper. We have carefully addressed your concerns on the technical challenge, technical contributions, and the comparison with FedAvg. Please let us know whether our answers accurately address your concerns. If our response resolves your concerns, we kindly ask you to consider raising the rating of our work. Thank you very much for your time and efforts! We would like to discuss any additional questions you may have.
>
> Best,
> Authors

---

### Official Review · Reviewer_rnQp · 2022-07-11

**Rating:** 5
**Confidence:** 4
**Soundness:** 2 fair
**Presentation:** 4 excellent
**Contribution:** 2 fair

**Summary:**

The authors provide a decentralized gradient clipping algorithm. The authors show that under some conditions the algorithm will converge to the $\epsilon$-stationary point with sample complexity $O(1/(N\epsilon^4))$ and communication complexity $O(1/\epsilon^3)$. And the algorithm performance shows the benefit of the proposed algorithm against naive parallel SGDclip.

**Questions:**

See weaknesses.

**Limitations:**

Yes

**Strengths And Weaknesses:**

Strengths:
1. Instead of using L-smoothness, the algorithm can converge under a weaker condition (L0,L1)-smoothness.

2. The experiments compare with the algorithm with wall clock time, which shows the speed up in the training phase.

Weaknesses:
1. With L-smoothness, [1] show the lower bound of communication iterations is lower bounded by $1/\epsilon^2$. But even when $L_1 = 0$, the number of communication iterations is still in the order of $1/\epsilon^3$.

2. (iv) in Assumption 1 is somehow strong. Even it can be shown in some layers for some experiments in Appendix E. Maybe change of Lemma 1 will be more important.

3. The hardest part of showing the result under the heterogeneous setting. in the homogenous setting, it seems that the most difficult part is dealing with I, the rest part will be the same as [51](ref in the paper).

4. The experiment would compare with more algorithms (e.g. clipping version of FedAVG)


Ref
[1] Zhang, X., Hong, M., Dhople, S., Yin, W., & Liu, Y. (2020). Fedpd: A federated learning framework with optimal rates and adaptivity to non-iid data. arXiv preprint arXiv:2005.11418.

---

> ### Author Response · Authors · 2022-07-29
> **Thank you for your comments. We have addressed your concerns accordingly.**
>
> **1. With L-smoothness, [1] shows that the lower bound of communication iterations is lower bounded by $1/\epsilon^2$. But when $L_1=0$, the number of communication iterations is still in the order of $1/\epsilon^3$.**
>
> > A: Thank you for mentioning this reference. We want to emphasize that the lower bound in [1] does not mean that our analysis is not tight. The reason is that the lower bound results in [1] require that every iterate is a linear function in terms of gradients, but our gradient clipping algorithm involves a nonlinear operator. So the lower bound in [1] does not apply to our algorithm since the clipping operator is involved, even assuming the function is $L$-smooth. We have cited [1] and discussed it in the revised version.
>
> > [1] Zhang, X., Hong, M., Dhople, S., Yin, W., & Liu, Y. (2020). Fedpd: A federated learning framework with optimal rates and adaptivity to non-iid data. arXiv preprint arXiv:2005.11418.
>
>
> **2. (iv) in Assumption 1 is somehow strong. Even it can be shown in some layers for some experiments in Appendix E. Maybe change of Lemma 1 will be more important.**
>
>
> >A: Please note that the same assumption is also made in the distributed learning literature (e.g., signSGD [Bernstein et al., ICML 2018]). More importantly, this is a reasonable assumption (especially because we empirically verified it in Appendix E). We believe this assumption helps us establish an important and generic technical lemma (Lemma 1 in Section 4.2) and it can inspire future research on distributed gradient clipping.
>
> > [Bernstein et al., ICML 2018] Jeremy Bernstein, Yu-Xiang Wang, Kamyar Azizzadenesheli, and Animashree Anandkumar. "signSGD: Compressed optimisation for non-convex problems." In International Conference on Machine Learning, pp. 560-569. PMLR, 2018.
>
>
>  **3. The hardest part of showing the result under the heterogeneous setting. in the homogenous setting, it seems that the most difficult part is dealing with I, the rest part will be the same as [51](ref in the paper).**
>
> > A: We respectfully disagree. Even the homogeneous setting proof is highly nontrivial, and we have highlighted the challenges in Section 4.1. First, we introduce a new Lemma (Lemma 1) for the truncated random variable which never appeared in the previous work to handle the difficulty of gradient clipping; Second, the rest of the proof is very different from [51] as well. The analysis strategy of [51] crucially requires global information to decide whether to do the clipping at every iteration, but our algorithm does not have that information so we have to do local clipping. To analyze the challenging setting of local clipping, we have to use a completely different analysis roadmap. Please check Section 4.4, where we describe the differences between ours and [51].
>
> **4. The experiment would compare with more algorithms (e.g. clipping version of FedAVG).**
>
> > A: We are not sure what the reviewer is referring to. We believe that our algorithm is indeed the clipping version of FedAvg. The contribution of this paper is to provide a novel and nontrivial analysis and thorough empirical verification for this algorithm.

---

> > ### Comment · Reviewer_rnQp · 2022-08-08
> > **Reponse to authors rebuttal**
> >
> > Thanks for the authors giving the explanation on all concerns.
> >
> > For concern 1, I agree that by introducing non-linear operator gradient clipping, the lower bound in [1] is no longer suitable to bound performance of the proposed algorithm. However, I'd like to know when the condition becomes gradient Lipschitz (i.e., $L_1 = 0$), whether the algorithm can perform as well as distributed SGD (in heterogeneous setting it only needs $\mathcal{O}(1/\epsilon^2)$ communication). If not, what is the main issue? (theoretical technique? or algorithm design?)
> >
> > For concerns 2 and 3, sorry for misunderstanding the contribution of Lemma 1. Then, from my understanding, for the technical part, the main contribution of this paper is giving a new lemma to deal with the bias introduced by gradient clipping. If this is correct, I think it should be added to the contribution of this paper. However, I still think that (iv) in Assumption 1 is strong even though it has been used in distributed learning.

---

> > > ### Author Response · Authors · 2022-08-08
> > > **Thank you for your response! Please check our responses below to address your further concerns.**
> > >
> > > Dear Reviewer rnQp,
> > >
> > > First, we would like to thank you for your response and are happy to address your further concerns.
> > >
> > > When the function is gradient Lipschitz with $L_1=0$, then our algorithm and analysis exactly recover local SGD by setting the clipping threshold $\gamma=+\infty$ and $\eta$ as the local SGD learning rate. However, without knowledge of $L_1$ (with the possibility that $L_1>0$), we have to use a more conservative stepsize and more frequent communication ($O(1/\epsilon^3)$ versus $O(1/\epsilon^2)$ in terms of communication rounds) to avoid gradient explosion. Please note that adapting to an unknown constant $L_1$ is known to be a difficult problem in stochastic nonconvex optimization. To the best of our knowledge, we are not aware of any work in distributed/federated learning literature, which is proved to be adaptive to smoothness constant without knowing it in advance.
> > >
> > >
> > > Regarding the (iv) assumption, we agree that it might be a bit strong. However, we want to kindly invite the reviewer to consider the following two points. First, this paper gives the first such analysis for the simple yet effective distributed gradient clipping algorithm with reduced communication in the relaxed smoothness setting; Second, this assumption was used in previous literature and is also verified empirically by our paper.
> > >
> > >
> > > In our humble opinion, science has to progress step by step. We believe that making the algorithm adaptive to $L_1$ or removing the assumption (iv) is important for future work. However, these tasks are typically extremely challenging. We believe that our work is still significant: it can be served as an initial investigation on the topic of distributed gradient clipping with reduced communication and linear speedup, and can inspire future research for further improvement.
> > >
> > > Best,
> > > Authors

---

> ### Author Response · Authors · 2022-08-08
> **Look forward to post-rebuttal feedback!**
>
> Dear Reviewer rnQp,
>
> Thank you for reviewing our paper. We have carefully answered your concerns on the lower bound, assumption, and main technical contribution. Please let us know if our answers accurately address your concerns. If our response resolves your concerns, we kindly ask you to consider raising the rating of our work. Thank you very much for your time and efforts! We would like to discuss any additional questions you may have.
>
> Best,
> Authors

---

### Official Review · Reviewer_ZjNg · 2022-07-11

**Rating:** 5
**Confidence:** 4
**Soundness:** 3 good
**Presentation:** 3 good
**Contribution:** 3 good

**Summary:**

This paper combines local sgd with gradient clipping. Local sgd is a communication-efficient method by skipping some aggregation rounds. Gradient clipping is a common technique to address the gradient exploding issue for some neural networks such as RNN and LSTM. By combining these two techniques, this paper achieves communication-efficiency and resolves the gradient exploding issue.

**Questions:**

In addition to my concerns as mentioned in the weaknesses above, I have the following questions:
* In Table 1 where the iteration and communication complexity of different methods are compared. Will the communication frequency I impact the communication complexity of the proposed method? I don’t see I in the table.
* In Figure 1a (right) where test accuracy vs epoch is shown, it looks like all methods (baseline and proposed methods with different I) converge to the same test accuracy. When I is very large (i.e., communication is less frequent), should the converged performance become worse? I noticed that same phenomenon in Figure 2a(right) and 3a(right).



**Ethics Review Area:**

["I don’t know"]

**Strengths And Weaknesses:**

Strengths:
* The motivation is clear. I think combining local sgd and gradient clipping makes sense.
* Theoretical analysis is provided. Especially, the communication complexity of the proposed method is proved to be better than the naive parallel method of the baseline gradient clipping.
* Experiments demonstrate that the proposed method is faster than the baseline in terms of wall-clock time in some cases.

Weaknesses:
* LSTM used to be the default method for processing sequential data such as in Natural Language Processing (NLP). However, now we can see the trend that transformer-based methods such as BERT has become the dominating method in NLP and related sequential data tasks. The benefits of transformer-based methods are obvious: easier parallelism leading to faster wall-clock time for training, and no need for back propagation via time (which would cause issues like gradient exploding). Given the current state-of-the-art established by transformer-based methods, I am afraid that the usefulness of the proposed method in this paper may be limited.
* Local SGD is primarily used in federated learning where the communication between clients and central server has a very high cost. I think it’s better if more experiments on federated learning (heterogeneous data distribution across clients, partial precipitation, etc) can be included.

==========after author response==========
I appreciate the authors' response. I understand there are some merits for the proposed method. I intend to keep my score unchanged mainly because I am still concerned with the usefulness of the proposed method due to the more recent transformer-based models as the current dominant methods in sequential data processing.

---

> ### Author Response · Authors · 2022-07-29
> **Thank you for your review. Please check our responses below.**
>
> **1. The benefits of transformer-based methods are obvious: easier parallelism leading to faster wall-clock time for training, and no need for back propagation via time (which would cause issues like gradient exploding). Given the current state-of-the-art established by transformer-based methods, I am afraid that the usefulness of the proposed method in this paper may be limited.**
>
> > A: Thank you for your insightful comments. Please note that our goal is to design provably efficient distributed algorithms with reduced communication rounds for a general class of relaxed smoothness functions. This is still a fundamental and open problem that is of interest to the distributed/federated learning community. We do not put the effort into studying transformer-based methods but we believe that it is not a weakness of our paper: we seek general algorithm design rules for a class of problems, which may also be applicable for transformer-based architectures. We plan to study this algorithm under the transformer framework in the future.
>
>
> **2. I think it’s better if more experiments on federated learning (heterogeneous data distribution across clients, partial precipitation, etc) can be included.**
>
> > A: Thank you for your suggestion. The partial participation experiment was included in Appendix D.4. In addition, we have also conducted heterogeneous data distribution experiments and included the results in Appendix D.9.
>
> **3. In Table 1 where the iteration and communication complexity of different methods are compared. Will the communication frequency I impact the communication complexity of the proposed method? I don’t see I in the table.**
>
> > A: $I$ is $T/R$, where $T$ is the number of iterations and $R$ is the communication rounds. We need $I\leq O(\frac{\sigma}{N\epsilon})$ to obtain linear speedup, which was already specified in the statement of Theorem 1.
>
> **4. In Figure 1a (right) where test accuracy vs epoch is shown, it looks like all methods (baseline and proposed methods with different I) converge to the same test accuracy. When I is very large (i.e., communication is less frequent), should the converged performance become worse? I noticed that same phenomenon in Figure 2a(right) and 3a(right).**
>
> > A: We would like to point out that we fine-tune each algorithm for the best training performance, in that the theory developed in this paper is meant for. We have run an experiment on training an AWD-LSTM on Penn Treebank with exponentially increasing $I$ and reported the results in Appendix Section D.10. There it can be clearly seen that as $I$ becomes large, the training performance gradually deteriorates. Nevertheless, we can observe that the convergence speed is almost the same when $I\leq 16$.

---

> ### Author Response · Authors · 2022-08-08
> **Look forward to post-rebuttal feedback!**
>
> Dear Reviewer zjNg,
>
> Thank you for spending the time to review our paper. We have carefully answered your questions, including the value of $I$ and new experiments with large $I$.
>
> Please let us know if our replies address your concerns. If our response resolves your concerns, we kindly ask you to consider raising the rating of our work. Thank you very much for your time and efforts! We would like to discuss any additional questions that the reviewer may have.
>
> Best,
> Authors

---

> ### Author Response · Authors · 2022-08-09
> **Could you please check our new experiments on imagenet?**
>
> Dear Reviewer zjNg,
>
> Thank you for your review. We know that you are a bit concerned about the lack of large-scale evaluation. We, therefore, worked hard to try our algorithm on ImageNet. It seems that local gradient clipping helps significantly on imagenet training in the Appendix D.14. Our algorithm spent only $44$ epochs and reaches $80\%$ validation accuracy, which is even better than a strong baseline trained with
> $90$ epochs [Goyal et al. 2018]: our algorithm is much better in terms of both epochs and wallclock time. Please check more details in the paper.
>
> [Goyal et al. 2018] Goyal, P., Dollár, P., Girshick, R.B., Noordhuis, P., Wesolowski, L., Kyrola, A., Tulloch, A., Jia, Y., & He, K. (2017). Accurate, Large Minibatch SGD: Training ImageNet in 1 Hour. ArXiv, abs/1706.02677.
>
> Hopefully, it can give you more confidence that our algorithm is useful in a large-scale experiment.
>
> Best,
> Authors

---

### Author Response · Authors · 2022-08-02
**General Comment**

We would like to thank all reviewers for their constructive comments. We have updated our manuscript according to the reviewers’ suggestions. All updates are marked in red. Please note that the majority of the changes are new experiments requested by Reviewers, which are included in the Appendix. The main summary of the changes are:

1. Per Reviewer ZjNg’s suggestion:

    * We have conducted heterogeneous data distribution experiments and included the results in Appendix D.9.

   * We have added experiments on our algorithm with exponentially increasing $I$ and reported the results in Appendix Section D.10.

2. Per Reviewer rnQp’s suggestion, we have cited and discussed the paper [ref1] in Section 2.

3. Per Reviewer xkbT’s suggestion, we have clarified that Appendix D.8 includes the comparison between FedAvg and our algorithm, and it indicates that clipping is crucial for good performance.

4. Per Reviewer ucfd’s suggestion:
    * We have added zoom-in figures (including Figure 1(a), 2(a), 3(a), 4, 6(a), 7, 9(a), 10(a), and 12) to make it easier for readers to differentiate performance of different methods.
    * We have summarized the results in tables in the Appendix including Tables 3, 4, 5.
    * We have added experiments on the momentum version of our algorithm in Appendix D.11.
    * We have run our algorithm multiple times and show the confidence interval, which is shown in Appendix D.12.

[ref1] Zhang, X., Hong, M., Dhople, S., Yin, W., & Liu, Y. (2020). Fedpd: A federated learning framework with optimal rates and adaptivity to non-iid data. arXiv preprint arXiv:2005.11418.

---

### Author Response · Authors · 2022-08-09
**New Large-scale Experiments Added: ImageNet Experiment**

Dear all Reviewers,

We would like to thank you all for your efforts in reviewing our paper. We have conducted imagenet experiments and added results in Appendix D.14. Our algorithm has got very good performance: our algorithm spent only $44$ epochs and reaches $80\\%$ validation accuracy, which is even better than a strong baseline trained with $90$ epochs [Goyal et al. 2018]: our algorithm is much better in terms of both epochs and wallclock time. Please check more details in the paper.

[Goyal et al. 2018] Goyal, P., Dollár, P., Girshick, R.B., Noordhuis, P., Wesolowski, L., Kyrola, A., Tulloch, A., Jia, Y., & He, K. (2017). Accurate, Large Minibatch SGD: Training ImageNet in 1 Hour. ArXiv, abs/1706.02677.

We hope it can clarify some of your concerns.

Best,
Authors

---

> ### Comment · Reviewer_ucfd · 2022-08-09
> **Follow-up ImageNet Experiments**
>
> Thank you for providing ImageNet experiments, the results are indeed impressive. I'm wondering where did the ~4% final accuracy gains came from compared to the baseline. Is it from simply using a gradient clipping scheme such as CELGC or is there something else hidden here. I think that these strong results should move up in the paper, and not hide in the appendix. Finally, listing some ImageNet wall-clock time speedup metrics such as time to accuracy would also strengthen the results.
>
> **Minors:**
> - The default weight-decay for ResNet-50 on ImageNet should be 1e-4 and not 5e-4.
> - The Naive Parallel SGDClip results seem to be cut off, I assume this was done due to time constraints, but I would like a confirmation.
> - Explanations are needed for the accuracy gains.

---

> > ### Author Response · Authors · 2022-08-09
> > **Thank you for your response!**
> >
> > Dear Reviewer ucfd,
> >
> > It is simply a gradient clipping scheme such as CELGC. We did not add any tricks except for applying warmup schemes in the first 5 epochs such as [Goyal et al. 2018]. We are also very excited about this result and will definitely move this result in the main text! We will also add wallclock time speedup results in the final version.
> >
> > For your minor questions:
> > 1. Yes. We will change the typo.
> > 2. Yes you are right. It takes lots of time to run: because it needs to communicate gradient and model at every iteration. We will include the results when it is finished. We expect that CELGC would be much faster than Naive Parallel SGDClip after training 90 epochs.
> > 3. We suspect the reason is that the clipping reduces the effective learning rate, which makes our algorithm more stable. We may explain this phenomenon by investigating the stability and generalization of stochastic algorithms such as (Theorem 3.8 in https://arxiv.org/abs/1509.01240). Smaller effective learning rate gives better stability bound which ends up with better generalization bound as well.
> >
> > Thank you very much for helping us improve our paper!
> >
> > Best,
> > Authors

---

### Meta-Review · Area_Chair_VX3y · 2022-08-26

**Recommendation:** Accept
**Confidence:** Certain

**Metareview:**

The paper presents a natural idea to combine local SGD with gradient clipping for communication efficient distributed training. Authors have addressed concerns from reviewers well (e.g., they included a ImageNet experiment). Overall well-motivated theoretical contribution with sufficient empirical validation. My recommendation is to accept this paper.

**Award:**

No

---

### Decision · Program_Chairs · 2022-09-14

Accept